# CRISPR-Cas12a for Highly Efficient and Marker-Free Targeted Integration in Human Pluripotent Stem Cells

**DOI:** 10.3390/ijms25020985

**Published:** 2024-01-12

**Authors:** Ruba Hammad, Jamal Alzubi, Manuel Rhiel, Kay O. Chmielewski, Laura Mosti, Julia Rositzka, Marcel Heugel, Jan Lawrenz, Valentina Pennucci, Birgitta Gläser, Judith Fischer, Axel Schambach, Thomas Moritz, Nico Lachmann, Tatjana I. Cornu, Claudio Mussolino, Richard Schäfer, Toni Cathomen

**Affiliations:** 1Institute for Transfusion Medicine and Gene Therapy, Medical Center—University of Freiburg, 79106 Freiburg, Germany; ruba.hammad@uniklinik-freiburg.de (R.H.); jamal.alzubi@uniklinik-freiburg.de (J.A.); manuel.rhiel@uniklinik-freiburg.de (M.R.); kay.chmielewski@uniklinik-freiburg.de (K.O.C.); mosti.laura@gmail.com (L.M.); julia.rositzka@uniklinik-freiburg.de (J.R.); marcel.heugel@live.de (M.H.); vale.pennucci@gmail.com (V.P.); tatjana.cornu@uniklinik-freiburg.de (T.I.C.); claudio.mussolino@uniklinik-freiburg.de (C.M.); richard.schaefer@uniklinik-freiburg.de (R.S.); 2Center for Chronic Immunodeficiency (CCI), Medical Center—University of Freiburg, 79106 Freiburg, Germany; 3Freiburg iPS Core Facility, Medical Center—University of Freiburg, 79106 Freiburg, Germany; 4PhD Program, Faculty of Biology, University of Freiburg, 79104 Freiburg, Germany; 5Institute of Human Genetics, Medical Center—University of Freiburg, 79106 Freiburg, Germany; birgitta.glaeser@uniklinik-freiburg.de (B.G.); judith.fischer@uniklinik-freiburg.de (J.F.); 6Faculty of Medicine, University of Freiburg, 79106 Freiburg, Germany; 7Institute for Experimental Hematology, Hannover Medical School, 30625 Hannover, Germany; schambach.axel@mh-hannover.de (A.S.); moritz.thomas@mh-hannover.de (T.M.); 8REBIRTH Center for Regenerative and Translational Medicine, Hannover Medical School, 30625 Hannover, Germany; lachmann.nico@mh-hannover.de; 9Department of Pediatric Pulmonology, Allergology and Neonatology, Hannover Medical School, 30625 Hannover, Germany; 10Biomedical Research in Endstage and Obstructive Lung Disease, German Center for Lung Research, Hannover Medical School, 30625 Hannover, Germany; 11Cluster of Excellence RESIST (EXC 2155), Hannover Medical School, 30625 Hannover, Germany; 12Fraunhofer Institute for Toxicology and Experimental Medicine (ITEM), 30625 Hannover, Germany

**Keywords:** genome editing, CRISPR, Cas12a, Cpf1, Cpf1 Ultra, iPSC, HSPC, HSC, T cell, AAV

## Abstract

The CRISPR-Cas12a platform has attracted interest in the genome editing community because the prototypical Acidaminococcus Cas12a generates a staggered DNA double-strand break upon binding to an AT-rich protospacer-adjacent motif (PAM, 5′-TTTV). The broad application of the platform in primary human cells was enabled by the development of an engineered version of the natural Cas12a protein, called Cas12a Ultra. In this study, we confirmed that CRISPR-Cas12a Ultra ribonucleoprotein complexes enabled allelic gene disruption frequencies of over 90% at multiple target sites in human T cells, hematopoietic stem and progenitor cells (HSPCs), and induced pluripotent stem cells (iPSCs). In addition, we demonstrated, for the first time, the efficient knock-in potential of the platform in human iPSCs and achieved targeted integration of a *GFP* marker gene into the *AAVS1* safe harbor site and a *CSF2RA* super-exon into *CSF2RA* in up to 90% of alleles without selection. Clonal analysis revealed bi-allelic integration in >50% of the screened iPSC clones without compromising their pluripotency and genomic integrity. Thus, in combination with the adeno-associated virus vector system, CRISPR-Cas12a Ultra provides a highly efficient genome editing platform for performing targeted knock-ins in human iPSCs.

## 1. Introduction

Genome editing is a major driver of progress in basic research, biotechnology, disease modeling and human gene therapy [1,2,3]. For targeted gene editing in preclinical and clinical applications, various classes of designer nucleases were developed, such as zinc finger nucleases (ZFNs), transcription activator-like effector nucleases (TALENs), and RNA-guided nucleases (RGNs). The prototypical example for the latter is the clustered regularly interspaced short palindromic repeat (CRISPR)/CRISPR-associated protein 9 (Cas9) system [4,5]. Precise gene editing following the introduction of a DNA double-strand break (DSB) at the intended target site is achieved by activating one of two main DNA repair pathways: non-homologous end-joining (NHEJ) or homology-directed repair (HDR) [6]. While all designer nucleases are composed of a programmable DNA recognition domain and a nuclease domain, they vary greatly in design. ZFNs and TALENs feature a protein-based DNA binding domain, whereas DNA recognition by RGNs is mediated by a guide RNA (gRNA) that is complementary to the target DNA sequence. The most commonly used CRISPR-Cas9 nuclease derives from Streptococcus pyogenes (SpCas9) [4,5]. For successful DNA cleavage, SpCas9 binds to a 5′-NRG trinucleotide, the protospacer-adjacent motif (PAM) [7], that flanks the target sequence (the protospacer).

Efficient genome editing in induced pluripotent stem cells (iPSCs) enables the validation of gene editing strategies in patient-derived cells and simplifies the setup of personalized disease models [8]. While NHEJ-based gene disruption approaches in pluripotent cells have been successful [9,10], HDR frequencies in iPSCs tended to be low when plasmid-based HDR templates were used [11,12]. Several publications have highlighted the reliance on marker-based selection strategies to enrich for edited iPSC clones, which is labor intensive, time consuming, and may have a negative impact on the pluripotency and the genome integrity of the selected clones [12,13]. Furthermore, many protocols still involve the use of feeder cells, which complicates the clinical translation of such cells and underlines the need to develop feeder-free and marker-free editing strategies. This became possible when CRISPR-Cas9 nucleases were combined with HDR templates based on adeno-associated virus (AAV) serotype 6 vectors, enabling precise genome editing at high frequencies [14,15].

To extend the targeting range of RGNs via the recognition of alternative PAMs, orthologous CRISPR-Cas9 [16,17,18] and CRISPR-Cas12a (also known as CRISPR-Cpf1) platforms [19,20,21] have been developed. Compared to the CRISPR-Cas9 system, Cas12a-based platforms have some distinct and attractive features: Cas12a enzymes are typically smaller in size than the SpCas9 nuclease, which simplifies vectorization, and the guide RNA, called CRISPR RNA (crRNA) in the context of Cas12a, is much shorter than a Cas9 gRNA [22]. In addition, the archetypal Cas12a derived from Acidaminococcus sp. BB3L6 (AsCas12a) or Lachnospiraceae bacterium ND2006 (LbCas12a) binds to AT-rich PAMs (5′-TTTV) and introduces staggered DNA cuts [23]. While these prototypical Cas12a nucleases had poor activity in primary human cells [23,24,25,26], an evolved variant of AsCas12a, termed Cas12a Ultra, has been shown to have high gene disruption capability in various human cell types, including iPSCs [9]. In addition, other Cas12a variants have been described, such as an engineered Cas12a nuclease isolated from Eubacterium rectale, termed ErCas12a or MAD7, which was shown to be active in human [27] and plant cells [28]. However, data demonstrating efficient Cas12a-mediated knock-in in iPSCs are still lacking in the field.

Here, we show that the Cas12a Ultra platform can mediate the targeted integration of DNA sequences in iPSC cells with high efficiency, obviating the necessity to apply selection schemes. To this end, we developed an antibiotic selection-free and feeder-free system to edit iPSCs with Cas12a Ultra. We compared the activity of the Cas12a Ultra platform to its wild-type counterpart AsCas12a in clinically relevant human cells, such as hematopoietic stem and progenitor cells (HSPCs) and iPSCs, and validated that the Cas12a Ultra variant can be used to efficiently target disease-relevant loci in these cells. In addition, to evaluate transgene knock-in strategies, we developed a GMP-compatible protocol to correct disease-causing mutations in iPSCs derived from patients with pulmonary alveolar proteinosis (PAP), a rare disorder characterized by an abnormal accumulation of surfactant-derived lipoprotein compounds within the alveoli of the lung. Overall, we demonstrated that Cas12a Ultra-based editing strategies in conjunction with AAV-based HDR templates are particularly effective in modifying the genomes of human iPSCs without compromising their pluripotency and genome integrity, paving the way for accelerating the discovery of novel curable, safe and effective therapies.

## 2. Results

### 2.1. Cas12a Ultra Efficiently Induces Gene Disruption in Primary Human Cells and iPSCs

To establish the Cas12a Ultra platform in our laboratory, we initially tested the gene editing potential of the Cas12a Ultra nuclease at several clinically relevant loci in primary human cells. The targets were *CSF2RA* coding for colony-stimulating factor (CSF) receptor alpha subunit, *CSF2RB* coding for CSF receptor beta chain, *CYBB* coding for gp91phox, *RAG1* coding for V(D)J-recombination-activating protein 1, *TRAC* encoding the T-cell receptor (TCR) alpha chain, and the safe harbor *AAVS1* (Figure 1A). We first determined the optimal Cas12a-Ultra-nuclease-to-crRNA ratio to achieve high gene disruption frequencies in human stem cells (HSPCs and iPSCs). To this end, we prepared RNPs to target the *RAG1* locus in HSPCs and the *CSF2RA* locus in iPSCs, respectively (Appendix A). The gene disruption efficiency was evaluated by assessing the extent of NHEJ-mediated mutagenic repair using the mismatched-sensitive T7 endonuclease 1 (T7E1) assay. We found that a 1:5 molar ratio of Cas12a Ultra protein to crRNA mediated the highest knockout efficiencies in both cell types with some 67% and 52% of *RAG1* and *CSF2RA* alleles being disrupted in HSPCs and iPSCs, respectively (Appendix A). Next, we verified that Cas12a Ultra outperforms the wild-type Cas12a in primary human cell types. Although active in vitro (Figure 1B), our data confirmed the absence of activity of wild-type Cas12a in cellula, regardless of cell type, genomic target, and nuclease concentration (Appendix A). In contrast, using Cas12a Ultra RNPs, we achieved NHEJ-based allelic gene disruption frequencies of up to 77%. Having identified the optimal conditions for Cas12a-Ultra-mediated gene disruption, we next compared the mutagenic activities of wild-type Cas12a and Cas12a Ultra across several loci in three different human cell types (Figure 1A). Our results confirmed high gene editing activity of Cas12a Ultra in iPSCs, HSPCs, and T cells, with frequencies ranging from 25% at the *CSF2RA* locus in iPSC to up to 90% editing of the *CYBB* locus in HSPCs (Figure 1C).

### 2.2. On- and Off-Target Activity of CRISPR-Cas12a Ultra

Next, we investigated the nature of CRISPR-Cas12a-Ultra-induced on- and off-target effects. Following nucleofection of the primary cells with the corresponding RNPs, the genomic DNA was extracted and targeted amplicon next-generation sequencing (NGS) was applied to assess the on-target activity. Our results confirmed high on-target activity at six different loci in iPSCs, HSPCs, and T cells, respectively (Figure 2A). Allelic editing frequencies ranged from 99% at *CSF2RA* in iPSCs and 89% at the *CYBB* gene in HSPCs to 27% at *AAVS1* and 64% at the *TRAC* locus in T cells, respectively. Targeted amplicon NGS at the eight highest-scoring in silico predicted off-target sites did not reveal signs of off-target activity.

In contrast to the preferential +1 insertions described for the Cas9 nuclease, Cas12a Ultra cleavage of genomic DNA preferentially resulted in deletions of 3–19 bp (Figure 2B). In general, we observed that the NGS-based assessment of the on-target activity detected higher indel frequencies than the T7E1 assay, e.g., evaluation of the *CSF2RA* gene disruption frequency via T7E1 resulted in ~25% of cleavage (Figure 1C), while >90% indels were detected via NGS (Figure 2A,B). This difference in the detection of indels implies that the T7E1 assay can severely underestimate knockout efficiencies, especially when indel frequencies exceed 50% and/or when specific types of indels, which are difficult to detect via the mismatch-sensitive T7E1 enzyme, predominate. To validate knockout activity at the phenotypic level, we targeted the TCR alpha chain encoding *TRAC* locus in primary T cells (Appendix A). Disruption of the *TRAC* locus reduced surface expression of the CD3/TCR complex by up to ~50% (Appendix A), which corresponded well with the allelic disruption frequency of 64% (Figure 2B). Increasing the amounts of CRISPR-Cas12a RNPs improved knockout efficiencies to up to ~60% without impacting cell viability (Appendix A). In brief, the CRISPR-Cas12a Ultra platform appears to be well suited for editing genes in primary human cells and iPSCs, with a trend toward more deletions than insertions.

### 2.3. CRISPR-Cas12a Ultra Mediates Effective Targeted Genomic Integration in iPSCs

Next, we sought to test the ability of the Cas12a Ultra platform to mediate the targeted knock-in (KI) of a transgene in iPSCs under GMP-compatible conditions. As a proof-of-concept, we targeted the integration of a GFP expression cassette into *AAVS1*. To deliver the GFP-encoding HDR template, it was packaged in a vector based on adeno-associated virus type 6 (AAV6) (Figure 3A). Then, iPSCs were nucleofected with RNPs targeting *AAVS1* followed by transduction with AAV6 carrying the HDR template. Upon nucleofection, the iPSCs were expanded for 13 days in the absence of selection before being evaluated for knock-in frequencies. Qualitative in–out PCR confirmed the targeted integration of the GFP template into the *AAVS1* site, as indicated by the detection of 5′- and 3′-junctions in the edited cells but not in the control samples (Figure 3B). Flow cytometric analysis revealed some 20% GFP-positive cells in samples edited with both Cas12a Ultra and AAV6 vectors, but not in the control samples (Figure 3C). While we observed initial toxicity in samples treated with both nuclease and AAV6 vectors at day 3 post-nucleofection/transduction, the edited iPSCs recovered and cell viability reached 80% at day 13 post-nucleofection/transduction for all samples (Figure 3D). Comparable knock-in frequencies were reached in primary T cells (Appendix A). Collectively, these results showed the ability of the Cas12a Ultra platform to mediate the targeted integration of an HDR template in human T cells and iPSCs at the polyclonal level.

Finally, we sought to correct the *CSF2RA* locus in an iPSC line derived from a patient with pulmonary alveolar proteinosis (PAP), a rare life-threatening lung disease with limited therapeutic options. It is characterized by an extensive accumulation of lipoprotein material in the alveolar space of the lung, ultimately resulting in respiratory failure and death [29,30]. To this end, a CRISPR-Cas12a nuclease targeting *CSF2RA* exon 3 and a corresponding HDR template coding for *CSF2RA* exons 3 to 13 were designed (Figure 4A). This *CSF2RA* transgene cassette, which was packaged into an AAV6 vector, would be able to restore the expression of the GM-CSF receptor in all cells that bear mutations downstream of the nuclease target site. Upon nucleofection of PAP-iPSCs with the Cas12a Ultra RNPs and transduction with the AAV6 HDR template, the edited cells were expanded without applying selection. Culturing the cells at a low density enabled the isolation of individual iPSC colonies for in-depth characterization. In–out PCR of individual clones that covered the 5′- and 3′-junctions confirmed the targeted integration of the HDR template into the *CSF2RA* locus in 11 out of 12 (92%) analyzed clones (Figure 4B,C). Allelic discrimination PCR was used to distinguish between mono-allelic and bi-allelic integration events. In 56% of the analyzed clones, both *CSF2RA* alleles were targeted, whereas 44% of colonies had a mono-allelic integration (Figure 4B,C). The sequencing of clone CSF2RA^KI#8^, which showed bi-allelic integration, confirmed the presence of the integrated super-exon (Appendix A). On the other hand, the genotyping of clone CSF2RA^KI#5^, with a confirmed mono-allelic integration of the super-exon, revealed a 7 bp-deletion on the second *CSF2RA* allele (Appendix A). Further characterization of clone CSF2RA^KI#8^ confirmed the absence of gross chromosomal aberrations (Figure 4D) and the preservation of pluripotency, as shown by the expression of SSEA4 and Oct4 (Figure 4E). In summary, Cas12a Ultra in combination with AAV6-based delivery of HDR templates constitutes a highly effective platform to edit iPSCs at the polyclonal level in a feeder-free, selection-free, and GMP-compatible manner without impairing the pluripotency or the genome integrity of the corrected iPSCs.

## 3. Discussion

Pluripotent stem-cell-based disease models are attractive tools to evaluate gene editing strategies using designer nucleases [8,11,12]. In the present study, we focused on validating a novel genome editing platform in iPSCs. To this end, we tested the potential of the CRISPR-Cas12a Ultra platform to edit various clinically relevant loci in primary human hematopoietic cells and iPSCs. A particular challenge of editing iPSC genomes is the need to select for edited clones and assess their quality, which is laborious, time consuming, and may even negatively impact the pluripotency and genome integrity of the selected iPSC clones. Manual selection of iPSC clones is considered the gold standard, especially following nuclease-based genome editing, to identify correctly engineered lines that harbor the desired mono- and/or bi-allelic editing outcomes [31,32]. The selected iPSC lines must then be carefully evaluated for pluripotency and genome integrity to avoid the selection of clones carrying undesired mutations [33,34,35]. It is therefore highly warranted to establish a platform that enables high-frequency editing without the need for marker-based selection or feeder cell-based culture.

Using a GMP-compatible setup for iPSCs, we achieved high editing activity, which resulted in allelic gene disruption frequencies of over 90%. The editing frequencies with the CRISPR-Cas12a Ultra nuclease in iPSCs and primary human T cells and HSPCs are therefore comparable to those achieved in clinically employed cells with the more established CRISPR-Cas9 platform [36,37,38]. For instance, in our case, using CRISPR-Cas12a Ultra to target the *TRAC* locus, we disrupted about 60% of *TRAC* alleles, which is comparable to the knockout efficiency achieved using CRISPR-Cas9 nuclease both in primary T cells [36]. Furthermore, we achieved ~90% knockout efficiency in HSPCs using Cas12a Ultra when targeting the *CYBB* locus, which is equivalent to the disruption frequency previously reported when using CRISPR-Cas9 nuclease in HSPCs [37]. These data are in stark contrast to our results and those of others [39] obtained with wild-type AsCas12a nuclease, which was active in vitro but not in cellula at any of the cells and loci tested. Consistent with previous results reporting a high specificity of the CRISPR-Cas12a Ultra system [9], our limited analysis found no evidence of off-target activity of this engineered nuclease. However, it should be noted that the amount of information on Cas12a off-target activity is nowhere near the extent of data available for Cas9. Recent studies attributed dsDNA nickase activity [40] and ssDNase activity to Cas12a [41]. Whether this is true for Cas12a Ultra remains to be determined. The effects of these Cas12a characteristics on the genome integrity certainly need further investigation [42]. It is worth noting that all loci targeted in this study harbor more potential SpCas9 target sites than AsCas12a target sites. This is, at least partially, owed to the fact that Cas12a Ultra recognizes a 4-bp long 5′-TTTV PAM in contrast to the shorter 5′-NRG PAM of SpCas9. On the other hand, because Cas12a binds to AT-rich PAMs rather than GC-rich PAMs recognized by SpCas9, it enables the targeting of sites in the genome that are not accessible to the canonical SpCas9 platform. Furthermore, due to its smaller size and the shorter crRNA when compared to CRISPR-SpCas9 [22], Cas12a Ultra is easier to vectorize and therefore a valuable addition to the genome engineering tool box.

Some interesting features of the Cas12a Ultra system, e.g., the nature of the induced DNA double-stranded breaks, are still underexplored. We found that Cas12a Ultra cleavage of genomic DNA does not lead to a preference for +1 insertions at the target site, as observed for the Cas9 nuclease, which may be due to the sticky nature of the DNA double-strand break ends induced by Cas12a as compared to the blunt ends induced by the Cas9 nuclease. Additionally, it would be worth investigating if staggered breaks with a protruding 5′-end promote HDR and therefore enhance the targeted integration of therapeutic cassettes in human cells. Li and colleagues described a microhomology-dependent integration strategy that enables a precise integration of DNA transgenes into desired loci in cell lines [43]. In a side-by-side comparison with the Cas9 system, they reported higher frequencies of precise integrations when using Cas12a over the Cas9 platform. To date, Cas12a-mediated gene editing has been described in yeast [44], some mammalian cell lines [43], and, recently, in primary human immune cells [9]. To the best of our knowledge, we show here for the first time that Cas12a Ultra is able to mediate targeted, bi-allelic integration of therapeutic cassettes in iPSCs at high frequencies without the need for antibiotic-based selection. HDR-based strategies are generally used to correct disease causing mutations or mediate the targeted integration of transgenes. The HDR machinery relies on the co-delivery of an HDR template that consists of a short single-stranded oligonucleotide, short double-stranded DNA, plasmid-based HDR templates, or AAV-based templates. For instance, in T cells, AAV6 vectors have been used to target the integration of a chimeric antigen receptor (CAR) into the *TRAC* locus [45]. AAV6-based HDR templates have also been utilized in HSPCs in the context of various preclinical disease models, including the successful restoration of immunodeficiency [46] and erythropoiesis [47]. The toxicity of AAV6 HDR templates in T cells appears to be tolerable, as indicated by the high knock-in frequencies without selection [48]. On the other hand, the use of this viral HDR template in HSPCs mediated high cytotoxicity, among others, through the activation of the p53 pathway, leading to impaired engraftment of these stem cells [49]. The most commonly used HDR template in iPSCs has been plasmid DNA, with sometimes striking negative effects on the editing efficacy, which necessitate the subsequent use of selection strategies to enrich for correctly gene-edited iPSC clones [12,13]. In our study, we demonstrate that AAV6-based HDR templates can be readily combined with the high activity of the Cas12a Ultra to correct disease-causing mutations in human iPSCs with high efficiency and without compromising the genome’s integrity. Our results hence encourage the broad application of Cas12a Ultra/AAV6-based genome editing strategies in pluripotent stem cells, e.g., for in vitro disease modeling, and the study of gene function and/or gene networks. Importantly, we have established a selection-free and feeder cell-free protocol to edit human iPSCs which is readily adaptable for clinical translation.

## 4. Materials and Methods

### 4.1. HSPC Isolation and Culture

Mobilized peripheral blood HSPCs were isolated from mobilized peripheral blood (IRB#329/10) using the CD34 MicroBead Kit UltraPure (Miltenyi Biotec, Bergisch-Gladbach, Germany, Cat# 130-100-453) according to the manufacturer’s instructions. Purified cells were resuspended in CryoStor CS10 (StemCell Technologies, Bothell, WA, USA, Cat# 07930) at 1 × 10^6^ cells/mL and stored in liquid nitrogen until usage. After thawing, and for 72 h prior to nucleofection, CD34+ cells were cultured in CellGenix GMP SCGM (CellGenix, Freiburg, Germany, Cat# 20802-0500) supplemented with recombinant human cytokines SCF (300 ng/mL), Flt3-L (300 ng/mL), IL-3 (60 ng/mL; all ImmunoTools, Friesoythe, Germany, Cat# 11343327, 11343307, 11340037), TPO (100 ng/mL; PeproTech, Hamburg, Germany, Cat# 300-18) as previously described [50]. After nucleofection (see section below), IL-3 was removed from the medium and HSPCs cultured at 37 °C with 5% CO_2_ until harvested for downstream analyses [50].

### 4.2. iPSC Generation and Culture

The iPSC lines were generated either from PBMCs of a PAP patient or from human foreskin fibroblasts derived from a healthy individual using Cytotune iPS 2.0 Sendai Reprogramming kit (ThermoFisher Scientific, Waltham, MA, USA, Cat#A16517). The line was characterized as previously described [51]. The PAP iPSC line was reprogramed from a patient harboring compound heterozygous mutations in exon 3 (A17G) and exon 7 (G196R). The iPSCs were cultured on matrigel- (Corning Life Sciences, Corning, NY, USA, Cat# 354277) or laminin 521 LN (Biolamina, Sundbyberg, Sweden, Cat# LN521)-coated plates under feeder-free conditions using mTesRTM1 Basal Medium (StemCell Technologies, Bothell, WA, USA, Cat# 85850) supplemented with mTesR TM1 supplement (StemCell Technologies, Bothell, WA, USA, Cat# 85852) at 37 °C with 5% CO_2_.

### 4.3. AAV Vector Production

HDR templates were cloned into plasmid pSUB201 [48,52] between the inverted terminal repeats (ITR) using NEBuilder HiFi DNA Assembly Master Mix (NEB, Frankfurt a. M., Germany, Cat# E2621). The relevant elements of the GFP HDR template consist of homology arms complementary to the *AAVS1* site (left 762 bp; right 735 bp), and a GFP transgene driven by the phosphoglycerate kinase (PGK) promoter followed by polyadenylation (pA) sequence derived from herpes simplex virus type 1 (HSVpA). The *CSF2RA* HDR template is composed of homology arms of 684 bp and 337 bp, respectively, which are complementary to the region surrounding the target site on exon 3 of the *CSF2RA* locus, a codon-optimized cDNA encoding exons 3 to 13 followed by a pA signal of bovine growth hormone (BGHpA). Recombinant AAV type 6 vector particles were produced and titers determined via ddPCR, as described previously [48].

### 4.4. Gene Editing of U2OS Cells

To form RNPs, 19 pmol of Cas12a Ultra (IDT, Coralville, ID, USA, Cat# 10001272) or wild-type Cas12a proteins (gift from IDT) and 95 pmol of indicated crRNA with 21 nucleotide protospacer sequence (IDT, Coralville, IA, USA) were complexed in a ratio of 1:5 before incubation for 10 min at room temperature (RT) (Table 1). U2OS cells (DSMZ, Braunschweig, Germany, Cat# ACC-785) were authenticated using a 16-DNA-marker profile (Eurofins Genomics, Ebersberg, Germany) and cultured in MEM low-glucose GlutaMAX (GIBCO/ThermoFisher, Waltham, MA, USA, Cat# 21885-25) medium supplemented with 10% FBS (PAN-Biotech, Aidenbach, Germany, Cat# P40-47500) and 1% penicillin/streptomycin (Sigma-Aldrich, St. Louis, MO, USA, Cat# P0781). Complexed RNPs were transferred to 4 × 105 cells via electroporation using program DN-100 and Kit SE (Lonza, Basel, Switzerland, Cat# V4XC-1032) on a 4D-Nucleofector (Lonza, Basel, Switzerland). After a 5 min recovery step, the cells were plated into 24-well plates for further use.

### 4.5. Gene Editing of Primary Human Cells

Gene editing in T cells and HSPCs was performed as previously described [48,50] with the following modifications: For T cells, RNPs at a 1:5 ratio (19 pmol of Cas12a and 95 pmol of crRNA) were transferred to 1 × 10^6^ T cells using program EO-115 and Kit P3 (Lonza, Basel, Switzerland, Cat# V4XP-3032) on a 4D-Nucleofector. For HSPCs, 2 × 10^5^ cells were mixed with Cas12a RNPs at a 1:5 ratio (38 pmol of Cas12a and 190 pmol of crRNA), and the RNPs transferred via electroporation (4D-Nucleofector, program CA137, Kit P3). For iPSCs, RNPs at a 1:5 ratio (19 pmol of Cas12a and 95 pmol of crRNA) were mixed with 4 × 10^5^ cells that were harvested at 70–80% cell density using TrypLE Select (GIBCO/ThermoFisher, Waltham, MA, USA, Cat# 12563011) before electroporation in a 4D-Nucleofector using program CA137 and Kit P3. After electroporation, the cells recovered for 5 min in iPSC medium supplemented with 10 µM Rock inhibitor Y-27632 2HCL (Seleckchem, Cologne, Germany, Cat# S1049) before being transferred to 24-well plates pre-coated with laminin 521 LN (see section above). Where specified, an overnight step at 32 °C was performed. For knock-in, the cells were transduced with AAV6 vector particles directly after nucleofection. The used vector doses were 3 × 10^4^ genome copies (GC)/cell for T cells and 1 × 10^4^ GC/cell for iPSCs.

### 4.6. Genotyping

T7 endonuclease 1 (T7E1; NEB, Frankfurt a. M., Germany, Cat# M0302L) assays were performed as previously described [53] (Table 2). For targeted amplicon sequencing, the target regions were amplified and the NGS library prepared and sequenced as previously described [36]. The generated paired-end reads were analyzed using the command line version of CRISPResso2 and R version 4.0.3 [54]. For the in silico prediction of potential off-target sites, Cas-OFFinder allowing up to 3 mismatches was used [55].

### 4.7. In Vitro Cleavage

In vitro cleavage was basically performed as previously described [56]. Briefly, purified PCR amplicons were incubated with CRISPR-Cas12 RNPs at a molar ratio of 1:4 (650 fmol of PCR amplicon with 2.6 pmol of Cas12 complexed to 7.8 pmol of crRNA) in NEBuffer 3 with BSA (NEB, Frankfurt a. M., Germany, Cat# B7003S) for 60 min at 37 °C. The reaction was stopped by adding 1 µL of STOP solution (30% glycerol, 1% SDS, 250 mM EDTA pH 8.0). The DNA was purified using AMPure XP beads (Beckman Coulter, Indianapolis, IN, USA, Cat# A63880) and then analyzed via agarose gel electrophoresis [56].

### 4.8. Cell Analysis

T cells were washed and stained with anti-human-CD3-APC antibody (Miltenyi Biotec, Bergisch-Gladbach, Germany, Cat# 130-113-125) in FACS buffer composed of PBS supplemented with 0.5% FBS and 2 mM EDTA (Serva, Heidelberg, Germany, Cat# 11280.02) at 4 °C in the dark for 20-30 min before data acquisition on a Accuri C6 flow cytometer (BD Biosciences, San Jose, CA, USA). FlowJo V10 (BD Biosciences, San Jose, CA, USA) was used for data analysis. Cell number and cell viability were determined using Solution 18 containing acridine orange and 4′,6-diamidino-2-phenylindole (DAPI) on a NucleoCounter NC-250 (ChemoMetec, Allerod, Denmark).

### 4.9. Statistical Analyses

GraphPad (GraphPad, San Diego, CA, USA) was used for data analysis and the creation of graphs. The number of repetitions performed for each experiment is indicated in the figure legend. Unless otherwise indicated, ANOVA tests (two-way, equal variance) were performed to determine statistically significant differences. *, **, and *** indicate *p* < 0.05, *p* < 0.01, and *p* < 0.001, respectively. The center values reported are the average, and the error bars represent the standard deviation (SD). Off-target frequencies were evaluated using CRISPRessoCompare [54]: *p*-values were adjusted using the Benjamini–Hochberg correction method [57].

## 5. Conclusions

The high activity of the Cas12a Ultra nuclease in human iPSCs and its broad range of putative applications, particularly in conjunction with AAV6-based HDR templates, are expected to have a significant impact on the field of iPSC-based therapies. Its ability to target AT-rich sites as well as its small size and short guide RNA will further contribute to its expanded use in pluripotent stem cells.

## Figures and Tables

**Figure 1 ijms-25-00985-f001:**
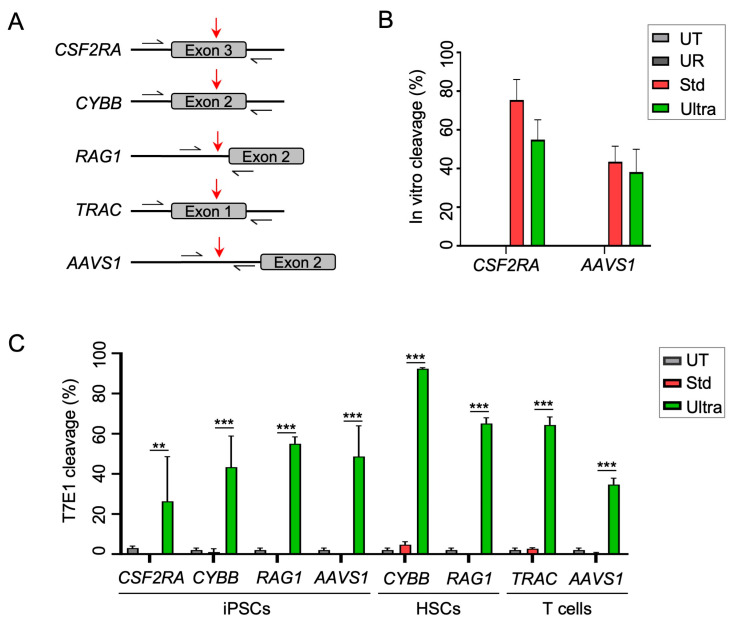
Genome editing in primary human cells. (**A**) Schematic of all target sites. Positions of primers used to amplify the targets for T7E1 assay are indicated by black arrows and cleavage sites by red arrows. (**B**) In vitro cleavage assay. Indicated targets were PCR amplified and subjected to in vitro cleavage with either standard Cas12a or Cas12a Ultra nuclease, respectively. Unrelated guided RNA (UR) and untreated target fragments (UT) were used as controls. (**C**) Genotypic analysis. Indicated primary cells were nucleofected with either Cas12a Ultra or standard Cas12a ribonucleoprotein (RNP) complexes. Knockout efficiencies were evaluated via T7E1 assay. UT, untreated; Std, standard Cas12a nuclease; Ultra, Cas12a Ultra nuclease. N = 3 independent experiments. ANOVA tests (two-way, equal variance) were performed to determine statistically significant differences. ** and *** indicate *p* < 0.01 and *p* < 0.001, respectively. The error bars represent the standard deviation (SD).

**Figure 2 ijms-25-00985-f002:**
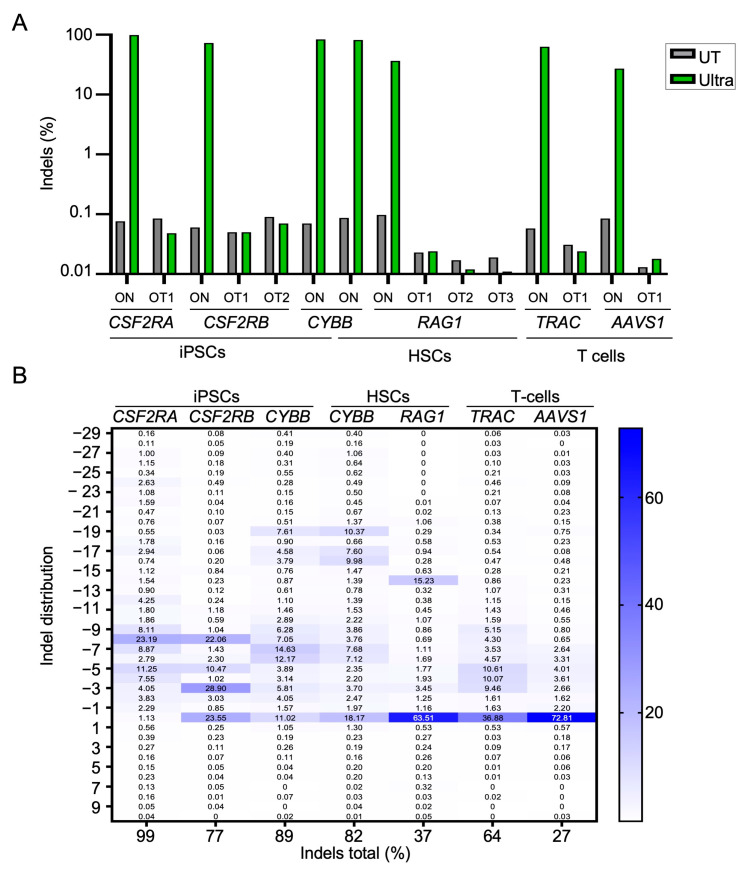
Gene editing outcomes. (**A**) Genotyping. On- and off-target activities in CRISPR-Cas12a-Ultra-edited human cells at indicated target loci were determined via NGS. Untreated cells (UT) served as controls. (**B**) Indel distribution. Indel distribution was determined in three human cell types for six different genomic loci. The table indicates the size and the percentages of insertions and deletions at the target sites. All values were normalized to the number of aligned reads.

**Figure 3 ijms-25-00985-f003:**
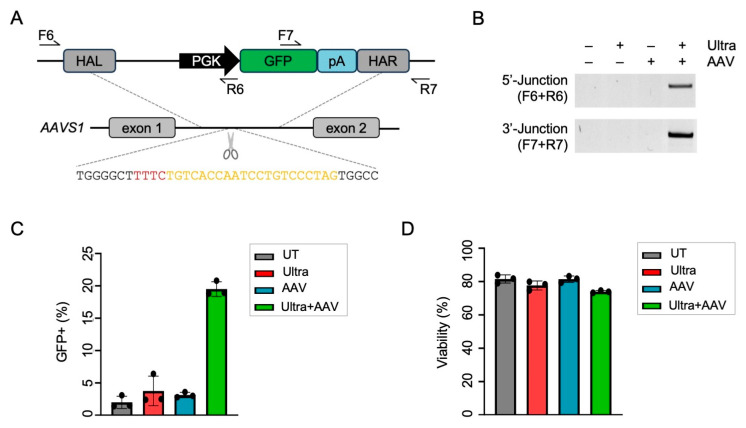
Knock-in of GFP into *AAVS1* in iPSCs. (**A**) Schematic of gene-targeting strategy. Shown are the *AAVS1* target site and the AAV6-based homology-directed repair (HDR) template. Relevant elements of the HDR template comprise a PGK promoter-driven GFP transgene followed by a Herpes simplex virus type-1 poly adenylation signal (pA), which is flanked by homology arms left (HAL; 762 bp) and right (HAR; 735 bp). Arrows indicate locations of primers used for genotyping. (**B**) Genotyping. iPSCs nucleofected with *AAVS1*-targeted CRISPR-Cas12a Ultra nuclease were transduced with the AAV6-based HDR template. Targeted integration was detected via 5′- and 3′-junction PCRs using the indicated primers. (**C**,**D**) Phenotyping. The percentage of GFP-positive iPSCs (**C**) and cell viability (**D**) was quantified via flow cytometry at day 9 post-transduction. N = 3 independent experiments. UT, untreated iPSCs; AAV; AAV6 only; Ultra, CRISPR-Cas12a Ultra only; Ultra + AAV, iPSCs nucleofected with CRISPR-Cas12a Ultra and transduced with AAVs HDR template; F, forward primer; R, reverse primer; PGK, phosphoglycerate kinase promoter.

**Figure 4 ijms-25-00985-f004:**
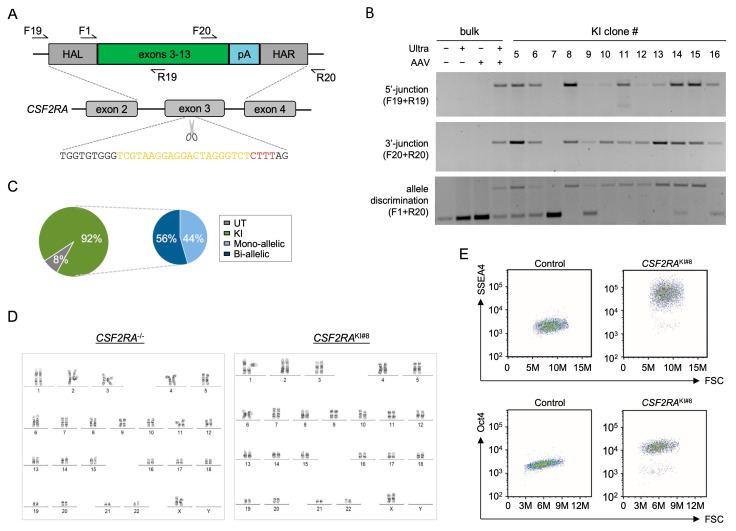
Targeted integration of a therapeutic *CSF2RA* super-exon in patient-derived iPSCs. (**A**) Schematic of gene-targeting strategy. Shown are the *CSF2RA* target site in exon 3 (bottom) and the AAV6-based homology-directed repair (HDR) template (top). Relevant elements of the HDR template comprise the homology arms left (HAL; 684 bp) and right (HAR; 337 bp) and the cDNA coding for *CSF2RA* exons 3-13 (super-exon) followed by the bovine growth hormone polyadenylation (pA) sequence. Arrows indicate locations of primers used for genotyping. (**B**) Genotyping. Patient-derived iPSCs nucleofected with *CSF2RA* targeted CRISPR-Cas12a Ultra nuclease were transduced with the AAV6 HDR template. Targeted integration was detected in individual iPSC clones (#5-16) and in the polyclonal cell population (bulk), and subjected to 5′-junction, 3′-junction and allelic discrimination PCRs using the indicated primers. (**C**) Quantification. Indicated is the fraction of iPSC clones with a targeted knock-in (green) as well as the percentages of mono-allelic vs. bi-allelic integration events (blue). (**D**) Genome integrity. Karyotyping was applied to iPSC clone#8 (CSF2RA^KI#8^) and compared to the patient-derived original iPSC clone (CSF2RA^−/−^). (**E**) Pluripotency. Corrected CSF2RA^KI#8^ clone was subjected to flow cytometry to assess the expression of pluripotency markers SSEA4 and OCT4. F, forward primer; R, reverse primer; UT, untreated cells; KI, knock-in.

**Table 1 ijms-25-00985-t001:** CRISPR-Cas12a target sequences.

crRNA Name	Target Sequence (5′-3′)
*CSF2RA*	TCTGGGATCAGGAGGAATGCT
*CSF2RB*	ACGGGGCCGCCGTGCTCAGCT
*CYBB*	TCTGGTATTACCGGGTTTATG
*RAG1*	CACATTGTATTAGCCTCATTG
*TRAC*	CACATGCAAAGTCAGATTTGT
*AAVS1*	TGTCACCAATCCTGTCCCTAG

**Table 2 ijms-25-00985-t002:** Primer sequence.

Primer Use	Target	Sequence (5′-3′)	Serial No.
T7E1 assay	*CSF2RA*-F1	tgtttccccaaatcacctcc	3982
	*CSF2RA*-R1	tcactacctggatggtcgttg	3983
	*CYBB*-F2	agtggcctgctatcagctac	4536
	*CYBB*-R2	atgtgtcactcctggatggattg	4310
	*RAG1*-F3	gttgcaggtttagagttccgtg	3363
	*RAG1*-R3	ggatctcacccggaacagct	3364
	*TRAC*-F4	taaagcatgagaccgtgact	3437
	*TRAC*-R4	tagacatcattgaccagagc	3438
	*AAVS1*-F5	ccttcttgtaggcctgcatcatcacc	1567
	*AAVS1*-R5	ggatcctctctggctccatcgtaag	1568
5′ and 3′ junction PCR	*AAVS1*-F6	ccagctcccatagctcagtctg	1207
	*AAVS1*-R6	gacgtgaagaatgtgcgaga	1405
	*AAVS1*-F7	ctacggcaagctgaccctgaa	154
	*AAVS1*-R7	gggctcagtctgaagagcagag	1208
	*CSF2RA*-F19	tcacgaggtcaggagatggag	5393
	*CSF2RA*-R19	aaggttcatagttcgactgtcg	5082
	*CSF2RA*-F20	acgatggcaacctcggttc	5394
	*CSF2RA*-R20	aggaacgaaggaacgaaggaac	5395
on-target analysis	*CSF2RA*-F8	tgccaggaatgtcctgggag	5042
	*CSF2RA*-R8	gggttggctggtgcttattgg	5043
	*CYBB*-F9	ctgactccagtcttgtgtggaatc	5044
	*CYBB*-R9	gagggagtgaggctaatggtac	5045
	*RAG1*-F10	caagtagtgaataattagtttctttgggtttgcagc	5046
	*RAG1*-R10	attcatctttgcctccccaagggt	5047
	*TRAC*-F11	acaagtctgtctgcctattcaccg	5048
	*TRAC*-R11	tcattgaccagagctctgggc	5049
	*AAVS1*-F12	ggcccctatgtccacttcag	5050
	*AAVS1*-R12	ctggcaaggagagagatggc	5051
	*CSF2RB*-F21	tgatgaatcacacggtgggc	6301
	*CSF2RB*-R21	gagagcaaggccaagaggag	6302
off-target analysis	*CSF2RA*-OT1-F13	cacagaagtcacatgctcctatattaaattctacg	5069
	*CSF2RA*-OT1-R13	gagggaacataagcccatttgtaaacagaaaatatga	5070
	*RAG1*-OT1-F14	tgcatacatgagcatccctaca	5071
	*RAG1*-OT1-R14	ctgtgacactacaggaaaagga	5072
	*RAG1*-OT2-F15	ggatttggctgctgcttctttctg	5073
	*RAG1*-OT2-R15	ctctgccaactaacttatgtcatgctatcaac	5074
	*RAG1*-OT3-F16	aagcaaggtccttgtgttgggg	5075
	*RAG1*-OT3-R16	cttctccttccagaaggggtc	5076
	*TRAC*-OT1-F17	ccatggcagtatagtggcacc	5077
	*TRAC*-OT1-R17	ctgcacctggcgtgtttttatcc	5078
	*AAVS1*-OT1-F18	tacagacacagcaggaggcc	5079
	*AAVS1*-OT1-R18	ctctcttgattctaccaccacattctactc	5080

## Data Availability

All data generated or analyzed during this study are included in this published article and its Appendix A files. Raw data, in particular targeted amplicon NGS data, have been deposited in the Gene Expression Omnibus repository and are available under GEO accession number GSE249731.

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
