# Peer review of "CRISPR-Cas12a for Highly Efficient and Marker-Free Targeted Integration in Human Pluripotent Stem Cells"

_ijms, 2024, doi:10.3390/ijms25020985_

Round 1
Reviewer 1 Report
Comments and Suggestions for Authors
Hammad et al. confirmed high NHEJ performance of CRISPR-Cas12a Ultra in human T cells, HSPCs, and iPSCs. They have also shown that Cas12a Ultra could mediate efficient knock-in of a GFP and a CSF2RA-expression cassette. The design and results of this study seems solid. However, the original Cas12a Ultra paper have displayed the great improvement of Cas12a Ultra compared to Cas12a, as demonstrated in HEK293T cell line and several human primary cells including the cell types used in this study, which largely reduces the novelty of this current study. This paper later showed that Cas12a Ultra can mediate impressive gene knock-in (>90% cell clones in the CSF2RA locus), which could be a very useful trait for homologous recombination users. But this has only been shown in 1 target locus, more validation would be strengthen this point.
Minor comments:
1. The authors showed great (>90% picked cell clones) knock-in of a therapeutic CSF2RA super-exon in iPSCs, but the knock-out (NHEJ) efficiency in Figure 1 is less than 30% on average. Did the authors use the same target in these two experiment? If so, how could you achieve such “high” HDR by a relatively “low” NHEJ frequency? Please explain.
2. Cas12a typically used crRNA, not gRNA. I could not find any info about the crRNA, e.g. how crRNA is made…
3. There are a few typos, e.g. 106 cells should be 106 cells. Please check the manuscript throughout.
Reviewer 2 Report
Comments and Suggestions for Authors
The manuscript by Hammad et al. takes a look at CRISPR-Cas12a Ultra an engineered version of As Cas12a. Much of the experimental data is similar to what has been shown during the original publication of the Cas12a Ultra with the exception that targeted integration was done in IPSCs using GMP-compatible protocols. However, the supporting data presented here with incremental improvements is sound and would likely be of interest to those considering use of Cas12a for genome engineering.
One thing that is particularly lacking is that this manuscript is framed by showing the poor activity of wild type As Cas12a versus the ultra version. But does not mention other Cas12a family members that seem to function better in than wild type As Cas12a. One example is ErCas12a or Mad7. These alternatives should be mentioned/referenced in the introduction so a naive reader is aware there are other related Cas family members that can be similarily used for targeted knock outs, editing, and inserting donor DNA.
The size of the homology arms (HAL and HAR) should be indicated in the figure or figure legend.
Overall this is a well written paper that demonstrates additional support for the increased activity of As Cas12a Ultra with some incremental advancements and demonstration of ISPC targeted integration.
Round 2
Reviewer 1 Report
Comments and Suggestions for Authors
The manuscript has been improved, but I have raised two points, which are marked in blue.
